# Tribological Properties of Additive Manufactured Materials for Energy Applications: A Review

Alessandro M. Ralls [1] , Pankaj Kumar [2,3] and Pradeep L. Menezes [1,*]

1   Department of Mechanical Engineering, University of Nevada Reno, Reno, NV 89501, USA; alessandroralls@nevada.unr.edu
2   Department of Chemical and Materials Engineering, University of Nevada Reno, Reno, NV 89501, USA; pkumar@unr.edu
3   Department of Mechanical Engineering, University of New Mexico, Albuquerque, NM 87106, USA
*   Correspondence: pmenezes@unr.edu

**Abstract:** Recently, additive manufacturing (AM) has gained much traction due to its processing advantages over traditional manufacturing methods. However, there are limited studies which focus on process optimization for surface quality of AM materials, which can dictate mechanical, thermal, and tribological performance. For example, in heat-transfer applications, increased surface quality is advantageous for reducing wear rates of vibrating tubes as well as increasing the heat-transfer rates of contacting systems. Although many post-processing and in situ manufacturing techniques are used in conjunction with AM techniques to improve surface quality, these processes are costly and time-consuming compared to optimized processing techniques. With improved as-built surface quality, particles tend to be better fused, which allows for greater wear resistance from contacting tube surfaces. Additionally, improved surface quality can reduce the entropy and exergy generated from flowing fluids, in turn increasing the thermodynamic efficiency of heat-transferring devices. This review aims to summarize the process-optimizing methods used in AM for metal-based heat exchangers and the importance of as-built surface quality to its performance and long-term energy conservation. The future directions and current challenges of this field will also be covered, with suggestions on how research in this topic can be improved.

**Keywords:** additive manufacturing; tribology; mechanical properties; metals; manufacturing; surface roughness; energy applications

## 1. Introduction

### 1.1. The Historical Development of Traditional Manufacturing to Additive Manufacturing

Based on the historical manufacturing review by Schmenner, he states that the concept of manufacturing was introduced during the industrial revolution in the sixteenth century [1]. This is largely supported from the historical development of modern-day industrialization, beginning with the rise of the United States from Britain during the 1700s. Understanding this fact, Hu has related this concept to the mass-production of consumer-based products in a review of the ever-changing paradigms of manufacturing processes [2]. Evolving from specialized paradigm type processes, larger, more "mass production" types of paradigms have emerged due to the invention of assembly lines. Serving the purpose of creating cheap and indistinguishable products, products such as the Ford Model T have popularized this concept which, in turn, has largely affected the manufacturing processes that we have today. More specifically, many fast and efficient manufacturing techniques have been adopted to fabricate more useful goods as opposed to previous practices, as identified by economist Alan Blinder [3]. This fact is supported by the increasing wealth of various countries within the past hundred years. With technology consistently improving, many innovative changes have occurred in manufacturing environments, as evident from the three industrial revolutions that have occurred.

Notable advancements in manufacturing processes were observed through the establishment of manufacturing factories in the late 1700s by Samuel Slater, as noted in his published biography by Conrad [4]. In fact, through the support of historians, Conrad has supported the fact that Slater has greatly contributed to the initial development of modern-day manufacturing through the invention of the water-powered cotton-spinning machine. In a separate biography of Slater, Tucker additionally supports this notion, including the idea that Slater is one of the catalysts of assembly-based processes through segmenting worker responsibilities to improve the quality and efficiency of various products [5]. Evidence of this was also supported by many referenced historians in this published biography. As consequence of this, Shah and Ward have identified that new manufacturing concepts such as "Just-In-Time Production" and "Lean Manufacturing" eventually arose, thus altering the perception of optimal manufacturing methods [6]. Although these concepts have mainly been recognized by the management-based literature, these approaches still have large importance for current-day manufacturing.

Through these novel concepts, additive manufacturing (AM) processes began to rapidly develop, dating back to the early 1900s, with the aim of manufacturing "Just-in-Time". One of the earliest AM systems proposed was the modern stereolithography technique, patented by Munz Otto in 1951 [7]. This system involved the formulation of photochemically etched objects through layers of photo emulsion. Based on the early invention of stereolithography techniques, the first commercial use of a stereolithography technique was in 1987 by a company named 3D Systems, naming their product the "SLA-1" [8]. 3D Systems described this product by using a ultraviolet-light-sensitive liquid polymer to create thin layers and building the product through layer-by-layer addition. Currently, this system is recognized in the National Inventors Hall of Fame Museum due to its profound impacts in the automotive and medical industries.

Following this invention, Kruth et al. describes the early 1990s as an era where many novel AM technologies, such as solid-ground-curing (SGC), laminated-object manufacturing (LOM), and fused deposition modeling (FDM), were developed and commercialized [9]. Throughout a decade span, Kruth has reported a ~34% increase in sales of these technologies from economical changes over time, which, at the time, was quite significant. That being said, although these novel processes used AM-based principles, many associate them with the concept of rapid prototyping (RP), this is a separate and distinct concept that should be understood. Horn and Harrysson describe RP as a concept that is more focused on the rapid timing of manufacturing prototypes compared to costlier and time-consuming subtractive-based methods [10]. Historically, the introduction of computer-aided designs (CAD) has drastically helped improve product-to-market time which, in turn, has popularized this manufacturing concept. Taking a macroscopic view of current-day manufacturing processes, there tends to be many influences from this concept, as markets are becoming much more competitive from the increasing expectation of customer demands. In contrast to this, Campbell et al. describes the concept of AM as the development of parts via layer-by-layer builds [11]. Although this concept may seem simplistic, its origins can be largely attributed to the evolution of RP-based processes, as explained by the authors of this review. The authors support this claim through the increasing uses of AM-based technologies in combination with their experience and observations throughout the years. That being said, although they may commonly overlap, it is emphasized that there is a distinct difference between the two processes. Based on both academic-based studies and commercial-based applications, Guo and Leu have identified the biomedical, aerospace, and automotive industries to have largely used AM-based processes since their development [12]. This can be supported through the increasing need to preserve material usage and conserve energy, as well as the need to reduce total weight while maximizing part performance.

In order to further visualize AM, this concept has the primary focus of developing three-dimensional components in a layer-by-layer fashion. Without the use of external tools such as cutting tools, coolants, or fixtures, AM allows for the user to limit the amount of

wasted material in a process, thus reducing total costs and allowing for a more conservative use of resources as opposed to traditional manufacturing practices.

As AM techniques continued to improve, many began to utilize metallic powders to achieve this novel building concept. This was first shown in a study by Ciraud where the metal powder was subject to the heat supplied by electron beams to create this layering concept [13]. As a consequence, this study acted as a building base for similar powder-laser processes in the future. Given the significance of this study, many modern-day AM-based processes would not exist without this research. With this novel concept gaining traction, the market for AM grew significantly in the early 2000s. This was especially evident in the period between 2010 to 2012, where the market grew by a total of 27.4%, demonstrating the potential of AM in diverse industries, as noted from Huang et al. [14]. In order to validate these trends, Huang et al. referenced the published findings from the well-established and reputable consulting firm Wohlers Associates, which has historically had profound impacts on the status of AM technologies worldwide.

### 1.2. Additive Manufacturing of Heat Transfer Devices for Energy Conserving Applications

Due to the enormous surge in AM technologies, many industries have welcomed their uses in order to further optimize manufactured components. Wong and Hernandez [15] provide a great insight to the many industries in which AM has been widely used in as well as their specific applications in their review of AM technologies. Common industries consist of the following: (1) aircraft, (2) aerospace, (3) architectural, (4) automotive, (5) biomedical, and (6) fuel-cell manufacturing, where their applications span from lightweight machining to architectural modeling and bone implant fabrication. However, in recent years, AM-based companies have extended their builds to help minimize the energy consumption of heat-exchanging devices. Examples of this are shown in studies conducted by Thompson et al. [16], Scheithauer et al. [17], and Cardon and Gargiulo [18].

Heat exchangers in their most basic form are devices that help transfer the thermal energy between multiple fluids. Through the generation of entropy, frictional effects are caused, which can impact the amount of energy consumed in an operation as well as the overall mechanical efficiency. By minimizing the amount of entropy generated in a process, the flow rate of the used fluid will be improved. Especially in large mechanical systems, the thermodynamic efficiency of heat-exchanging devices will largely impact the parts' systems performance [19]. As consequence, additive manufacturing has been proposed as one method which can improve the efficiency of AM heat exchangers. Through the distinctive and intricate designs offered by AM, heat exchangers can obtain higher efficiencies as the surface-to-volume ratio of components is optimized [20]. In addition to this, the complex metal matrix composites (MMC) can be used in AM laser-based fabrication, which can help alter the heat-transferring coefficient present with the build. However, one issue that may arise from AM heat exchangers is the control of as-built surface roughness. With a large surface roughness, heat-transferring factors such as convective heat transfer may be negatively impacted due to the non-optimized heat transfer rate [21]. In addition to this, rougher surfaces have been shown to increase the turbulent kinetic energy of the flowing fluid, which impacts its flow in a large system, with greater material wear rates over time [22]. Especially with contacting surfaces, tube materials can be preserved for longer periods of time. Control over the as-built surface roughness is also a common issue in general additive manufacturing research, where surface roughness induced from laser-melting negatively affects the parts' mechanical properties [23].

To combat this, researchers have studied smoothing techniques which can be applied after the manufacturing process [24]. From an enhanced surface finish, the mechanical strength of the build will be improved from a fatigue [25], wear, and mechanical [26] point of view. However, one downside from post-surface finishing techniques is the time and resources required to purchase, run, and maintain these machines. Especially from a time efficiency standpoint, it may take longer periods of time to smoothen larger or more complex surface geometries such as heat exchangers. As consequence, there is a need to

enhance many aspects of the as-built surface roughness. By reducing the surface roughness, the material will benefit from the following aspects: (1) indication of the melt quality of the build [27], (2) densifying the build [27], (3) decreasing porosity, (4) improving fluid flow [22], (5) increasing mechanical/friction/wear properties [25,26], and (6) reducing the heat loss from transferring fluid [21]. By improving all of these aspects, heat-exchanging devices will consume much less energy and have higher efficiency rates than traditional heat-exchanging devices. Klien et al. [28] have also recognized these gaps in research in their review of additively manufactured heat exchangers; however, a comprehensive review which establishes the relationship between surface roughness, mechanical/tribological performance, and their impacts on heat-exchanging devices spanning from part-build to in situ performance, does not exist to date. Many have studied each of these topics separately (Table 1); however, this research is needed in order to advance the development of additive manufactured heat exchanges. This review will aim to encapsulate the current literature on AM metals and the relationship between as-built surface finish and mechanical/tribological performances—more specifically, individual research from surface-quality optimization for AM builds, their effects on their mechanical–tribological properties, and their impacts on the performance of heat-exchanging devices. Using this information, their applications towards enhancing the quality of heat exchangers will be discussed, with a conclusion about the topic of this research and how it can progress moving forward.

**Table 1.** Publications relating to mechanical, tribological, metrological, and energy preserving enhancements of additive manufacturing (AM) metals.

| Article Name | Summarization | Reference |
|---|---|---|
| Metal Additive Manufacturing: A Review | • General manufacturing systems are highlighted<br>• Technological challenges are highlighted.<br>• Mechanical properties of TI-6Al-4V, IN 625, and IN 718 are investigated.<br>• Business and environmental considerations for AM metals are discussed. | [29] |
| The Metallurgy and Processing Science of Metal Additive Manufacturing | • Techniques for manufacturing metal-based parts are investigated.<br>• General material processing issues for AM metals are explored.<br>• Solidification, heat transfer, mechanical properties, post-processing metallurgy, processing defects, and solid-state precipitation for select AM metals are studied. | [30] |
| Metal Additive Manufacturing: A Review of Mechanical Properties | • The general AM techniques discussed included directed energy deposition (DED) and powder bed fusion (PBF)-based processes.<br>• Majority of data are derived from studies using Ti-6Al-4V.<br>• Quasi-static properties are highlighted and discussed.<br>• Problems regarding high cycle fatigue and crack growth regarding surface quality is discussed. | [31] |
| Surface Texture Metrology for Additive Manufacturing: A Review | • Metrology literature regarding AM metals is reviewed.<br>• Surface measurement and characterization technologies are investigated.<br>• Industrial applications are investigated.<br>• Overview of general metals used in metrological-based AM is discussed as well as their applications in industry. | [27] |

**Table 1.** *Cont.*

| Article Name | Summarization | Reference |
|---|---|---|
| Additive manufacturing of Ti-6Al-4V alloy: A review | • Progression of AM Ti-6Al-4V is reviewed.<br>• Laser direct deposition (LDD), electron beam melting (EBM), and selective laser melting (SLM)-based processes were mainly discussed.<br>• Fatigue and tensile properties are investigated.<br>• Influences of defects from built components are discussed. | [32] |
| Additive manufacturing of metallic components by selective electron beam melting—A review | • Selective electron beam melting (SEBM)-based metals are reviewed in the following study.<br>• Relationships between processing parameters and material properties are discussed.<br>• Co-Cr-Mo-, Cu-, Fe-, Nb-, Ni-, Ti-, and TiAl-based materials were investigated. | [33] |
| Effects of Process Parameters on the Surface Roughness and Mechanical Performance of Additively Manufactured Alloy 718 | • Processing optimization of AM nickel-based superalloy 718 is studied.<br>• Surface roughness and its relation to mechanical properties are listed.<br>• Finite element is used for experimentation in this study. | [34] |
| A Review of Recent Advances in Additively Manufactured Heat Exchangers | • Advantages of AM heat exchangers are discussed.<br>• Most up-to-date information pertaining to the performance of part geometry is discussed.<br>• Metals, polymers, and ceramics are discussed. | [28] |
| Potentials and Challenges for Additive Manufacturing Technologies for Heat Exchanger | • AM for complex heat-exchanging geometries is discussed.<br>• The advantages for AM for designs such as bent structures are highlighted.<br>• The flow performance affected by AM surface roughness is discussed. | [35] |

### 1.3. The Importance of Tribological Property Alterations in Additive Manufacturing

As explained in the book Tribology for Scientists and Engineers, tribology is defined as the science of studying friction and wear from two contacting surfaces [36]. Through the accumulation of references from published studies from Menezes et al., this work acts as a building base for the concepts that will be discussed in this review. With a scope ranging from the topic of tribology to AM, tribological properties have a tremendous impact on part performance and can dictate whether a fabricated part will succeed or fail in crucial high-volume industries such as aerospace, transportation, and automotive. This is especially the case in energy-consuming parts, where friction and wear are large indicators of how much energy is being used in an application. With friction and wear being system characteristics, one way of optimizing tribological performance is to manipulate the intrinsic properties of the selected system through the average grain size.

With friction and wear being closely related to the mechanical properties of a system, Yan et al. [37] are among many researchers who have established a link between grain size to friction and wear. In this study, various building directions of SLM'd Inconel 625 studied were used: 0°, 45°, and 90°. Upon investigation, the 0° build showed to have the largest quantity of equiaxed grains, with the average grain size being the smallest out of the other two build directions. The authors largely attribute this phenomenon to the distribution of thermal energy from the laser. As the laser increased in radial distance (i.e., scan angle), there was shown to be a decrease in convective heat transfer throughout the melt zone,

which, in turn, negatively impacted the grain size. As the grain size decreased, both the mechanical and tribological strength were drastically increased, as supported by the Hall–Petch relationship. Understanding that this phenomenon is widely known and time-tested, it is evident that the friction and wear properties of a system do indeed share an inverse relationship, which can be greatly beneficial in energy-related applications.

However, tribological performance can also be highly correlated with the finishing surface texture of metal-based components. Costa and Hutchings [38] support this in their review on the impacts of surface texturing techniques for tribological optimization. Specifically, the authors of this study gave many examples of the literature that has noted an increase in the tribological performance of metals that have specific surface textures. This is especially true in lubricate-based conditions, where textured dimples can help minimize third body wear by storing any wear debris inside of the pockets. Given that the textured system is subjected to some wear rate overtime, its life cycle would also be quite dependent on the surface features of a part, as this helps increase its total quality, energy-conserving effects, and long-term performance. Even in non-textured conditions, lubrication plays an impactful role in energy conservation through reduced wear-rates.

Acting as a substance that improves the efficiency and performance of mechanical systems, lubricants can be broken down into various classifications depending on their state of matter. The most popular forms of lubricants come in the form of liquids, solids, or gases. In applications where the lubricant is continually circulated, such as in bearings, liquid lubricants are more preferred due to their viscous nature [36]. Likewise, in applications where the lubricants are exposed in extreme conditions (e.g., elevated temperatures), solid lubricants such as molybdenum disulphide ($MoS_2$) are more suitable due their controlled viscosity in contrast to liquid lubricants [39,40]. In contrast with the two previous forms of lubricants, gaseous lubrications are generally sought out due to their low viscous nature. Generally, gaseous lubrication is utilized in bearings in high-speed machineries. It should be mentioned that in the lubrication literature, additives are often utilized which, in turn, can promote enhanced heat transfer rates and an increase in material preservation [36].

Even with the various types of lubrication, the function of lubrication can be further classified into four regimes bounded by the Stribeck curve. These regimes are classified as (1) boundary lubrication, (2) mixed film lubrication, (3) elastohydrodynamic lubrication, and (4) hydrodynamic lubrication [36]. Investigating the first regime, boundary lubrication describes the lubrication that takes place between two contacting surfaces. Often quite complex, boundary lubrication can be influenced by a multitude of factors including the friction force present in the system, the velocity, and the type of lubrication utilized. Greenfield and Ohtani [41] provide an insightful view on the friction and normal forces that occur during boundary lubrication as the sliding velocity and surface separation of the lubricant is changed. In instances where there were no trapped layers, the friction was recorded to be the highest due to the interatomic separation of the liquid being constant, thus implying little to no variations in film thickness. Lin and Klien [42] have also contributed to this research in their review of surface forces being controlled during boundary lubrication. Following boundary lubrication, mixed-film lubrication is a hybrid combination of boundary lubrication and hydrodynamic lubrication. Translating this concept to a sliding application, this lubrication regime consists of some contacts from the surface asperities; however, there is still a film of lubrication preventing full contact between the asperities from both contacting surfaces. Transitioning into elastohydrodynamic lubrication, this regime of lubrication is dependent on the elastic deformation that occurs from the contacting asperities, somewhat similar to the mixed lubrication regime [36]. The last lubrication regime, hydrodynamic lubrication, is quite different from the previous two regimes due to the generation of hydrodynamic pressure occurring between two moving surfaces. Generally, the film of fluid from the lubrication is thick enough to prevent any interaction from the two moving surfaces, which can be greatly advantageous for mechanical efficiency, as highlighted by Heli et al. [43].

In a global sense, through improving the tribology of contacting surfaces, Masjuki et al. [44] estimated that 40% of the energy loses via friction and wear will be able to be reduced by 2032, based on the advancements and trends of tribological technologies. Maskuji et al. supports this claim through calculating the energy consumption of heavy-based vehicles and correlating the energy losses of these vehicles from friction. These calculations were based on two published reviews focusing on recent reviews of the influence of tribology on global energy consumption. Expanding on one of these references, Holberg and Erdemir [45] have reported that ~23% of the global energy is consumed by tribology. These calculations were done by evaluating the most recent reports on the global supply of energy and calculating their tribological energy losses by industrial sectors. Given the number of citations this paper has, as well as the amount of referenced material, it is safe to assume that these estimates are quite accurate. Nonetheless, the main takeaway from these studies is that AM-based industries can greatly benefit from enhanced surface-based properties, as they will enhance part performance, manufacturing costs, and long-term energy conservation.

When evaluating the tribological performance of manufactured components in relation to part quality, part performance critically depends on the surface of the part. In general, various surface characteristics are used to describe this. Through evaluating the surface, manufacturers can make three different evaluations about the manufactured part. First, the surface features of a part can indicate the general tolerance of complex components, and whether they can be assembled together accurately or not. Although this may seem like a basic and intuitive point, authors such as Leach [46] emphasize this due to its sheer importance from an industrial standpoint. Leach elaborates on this point by giving examples of aerospace-based parts such as turbine blades, and how manufacturers would not fly these blades without having extreme confidence in their metrological properties. With larger-sized clearances, post-processing processes such as near-net-shape manufacturing may be needed in order to minimize any material wastage. The optimization of having an accurately fitting part can have profound impacts on specific fit-for-purpose operations. One example can be shown in Donoghue [47] and his investigation of hybrid additive manufacturing techniques for aerospace-based components. In his example, he elaborates on the use of bearings for aerospace-based applications. In the event that the frictional heat of the bearing resulted in early part failure, having a proper and accurate clearance would allow for additives such as lubricants to enter, and allow for an improvement in friction. Second, having an enhanced surface finish will minimize the amount of post-processing processes needed to smooth the surface. This, in turn, improves the efficient use of energy and time in additive manufacturing processes.

Lastly, and most relevant to this paper, a high-quality surface finish will indicate how successfully the particles have been fused in the AM process. If the metallic particles are successfully melted, there will be a lower chance of pores and impurities on the surface. In the case that surface quality is enhanced, there will be a lower chance of any cracks being formed when an external force is applied, thus enhancing the mechanical and tribological properties of the system. Zhu et al. [48] were able to find this through their publication focusing on the friction and wear performance of 316L fabricated from selective laser melting (SLM). With the presence of micropores, the molten pools began forming a spheroid shape which, in turn, roughened the surface and negatively impacted the tribological performance of the part. Likewise, Kovacı and Seçer [49] have also noted an increase in cracking of SLM 316L stainless steel due to the presence of voids and pores from the force induced from their pin-on-disc tests. Although surface texturing techniques may be used to mitigate these issues, if there is a large number of voids present on the surface, the change in geometry from texturing may not be as effective compared to a scenario where there are fewer general voids on the material surface.

In order to quantify and compare the surface features of additive manufactured parts, many three-dimensional surface standards have been created by the International Organization for Standardization (ISO) [50]. Being a non-governmental organization, ISO

holds a very prestigious reputation, seeking to create international standards for a wide spread of industries. Although prior surface measurements were created by Birmingham University in 1990, the parameters based on the Geometric Product Specification (GSP) for surface texturing were primarily to describe surface features [51]. Despite this, ISO had created more established parameters that would serve to set the standards for areal surface texturing. Generally, ISO 25178 and ISO 4287-1 are the two standards that are referred to in the literature. The differentiating factor between these two standards is the method of surface measurement. With ISO 4287 [52], these measurements are based off a stylus-based profile measurement. For ISO 25178, the measurements are supported in uses for both contact and non-contact uses [53]. These parameters account for the infinite number of variations in amplitude (vertical) and spacing (horizontal), and their combination in terms of peaks, valleys, slopes, and all other geometrical designs of the surface. Examples of commonly used parameters are described in Table 2.

**Table 2.** General surface roughness parameters from ISO 25178 [53], ISO 4287 [52], and from Tribology for Scientists and Engineers [36].

| Amplitude Roughness Parameters | | | |
|---|---|---|---|
| $R_a$: Average Roughness | $R_a = \frac{1}{n}\sum_{i=1}^{n}|z_i - \bar{z}|$ | • $n$ indicates the total number of data points of the surface profile. <br> • $z_i$ and $\bar{z}$ are the data points on the surface profile and its average, respectively. | Average deviations in the vertical asperities are measured. |
| $R_q$: Root-mean-square roughness | $R_q = \sqrt{\frac{1}{n}\sum_{i=1}^{n}(z_i - \bar{z})^2}$ | • $n$ indicates the total number of data points of the surface profile. <br> • $z_i$ and $\bar{z}$ are the data points on the surface profile and its average, respectively. | Standard deviation in vertical asperities is measured. |
| $R_{sk}$: Skewness | $R_{sk} = \frac{1}{nR_q^3}\sum_{i=1}^{n}|z_i - \bar{z}|$ | • $n$ indicates the total number of data points of the surface profile. <br> • $z_i$ and $\bar{z}$ are the data points on the surface profile and its average, respectively. <br> • $R_q$ is the standard deviation of the asperitiy heights. | The total skewness of the surface asperities is described. Skewness is generally shown to be either negative or positive. |
| Spacing Parameters | | | |
| $P_c$: Peak Count | $P_c$ | • $P_c$ indicates the number of peaks on the material surface | The number of local peaks is described. |
| $S$: The Median Spacing on the Meal Line | $S = \frac{1}{n}\sum_{i=1}^{n}S_i$ | • $n$ indicates the total number of data points of the surface profile <br> • $S_i$ indicates the difference in horizontal measurement of local peaks | The average spacing between all local peaks is described. |
| $S_m$: Average Spacing of Adjacent Local Peaks | $S_m = \frac{1}{n}\sum_{i=1}^{n}S_i$ | • $n$ indicates the total number of data points of the surface profile <br> • $S_i$ indicates the difference in horizontal measurement of local peaks | The average spacing between local peaks above the average boundary line is described. |

**Table 2.** *Cont.*

| Amplitude Roughness Parameters | | | |
|---|---|---|---|
| **Hybrid Parameters** | | | |
| $\Delta_a$: Mean Slope | $\Delta_a = \frac{1}{n-1}\sum\limits_{i=1}^{n-1}\frac{\delta_{yi}}{\delta_{xi}}$ | • $n$ indicates the total number of data points of the surface profile<br>• $\frac{\delta_{yi}}{\delta_{xi}}$ describes the slopes between two points | Mean slope aims to describe the average measurement slopes between all points |
| $\lambda_a$: Mean Wavelength | $\lambda_a = \frac{2\pi R_a}{\Delta_a}$ | • $R_a$ is the average deviation in the vertical asperities<br>• $\Delta_a$ is the mean value of all the slopes on the surface | This value measures the spacing between all valleys and peaks |
| $\lambda_q$: Root-mean-square of Wavelength | $\lambda_q = \frac{2\pi R_q}{\Delta_q}$ | • $R_q$ is the standard measured deviation in vertical asperities.<br>• $\Delta_q$ is the root–mean–square for the average slope measurement of the profile | This value measures the standard deviation in all wavelengths on the surface. |

*1.4. Impacts of Surface Roughness on Mechanical and Tribological Properties for Additive Manufacturing Practices*

In a general sense, the vast array of metrological parameters tends to have their own specific impacts on the surface properties of AM'd metals. Surface texturing is one method that allows for the manipulation of surface parameters, which can improve part quality and prevent many defects such as balling effects and Marangoni circulation from occurring. One example of this can be found in a surface texturing review from Townesend et al., where it is stated that evaluating surface metrology acts as a method to help understand any physical phenomena that occur as a byproduct of AM [27]. Given the examples of balling-based effects and unoptimized heat transfer, it makes sense that AM-processing variables can impact the surface properties of metal-based parts. This is especially true in cases where the part is exposed to contacting surfaces. As a general example, Bai et al. [54] were able to find a correlation between skewness ($R_{sk}$) and wear loss for lubricated 52,100 steel. The lubricant used in this study is molybdenum dialkyldithiophosphate, also known as MoDDP. This lubricant is widely known to reduce friction for automotive-based applications through the formation of a tribofilm. As $R_{sk}$ was reduced, the deeper valleys on the material surface allowed for greater storage of lubrication and wear debris from the broken asperities. As a consequence, the wear rate of the steel showed great improvements over time. In addition to this, general part properties such as friction, wear, and general mechanical properties can be enhanced, as they depend on contact-based energy operations.

Cabanettes et al. [55] are among the many researchers who have created different links between AM-processing variables and select surface roughness parameters. In their study, the surface properties of SLM Ti-6Al-4V were investigated as the inclination angle of the fabricated plates varied. The inclination angles were varied between positive "upskin" angles of $\alpha$ = 0–90° (in increments of 10°) and negative "downskin" angles of $\alpha$ = 50–90°. Given that laser power (300 W), laser scan speed (1800 mm s$^{-1}$), hatch space (85 µm), beam size (70 µm), and layer thickness (30 µm) remained constant, various amplitude, spatial, and hybrid parameters were studied. Most notably, changes in skewness ($S_{sk}$) in relation to the number of peaks ($S_{pk}$) and valleys ($S_{vk}$) were highlighted. In tests with an upskin deposition angle, there was an inverse relationship between an increase in peaks and fewer valleys, which, in turn, resulted in a positive skewness. In contrast to these results, the opposite was shown for downskin-based tests. The relationship between downskin and upskin angles to skewness is shown in Figure 1. Based on the other literature [56], having a lower skewness rating generally results in lower surface friction, even if the general surface roughness tends to be high. Although these relations were derived from 100Cr6 steel, we

can make sense of this through the definition of skewness, as having a smaller value of skewness means that there are fewer peaks than valleys in the studied surface, which can be quite helpful in low-friction-based energy operations.

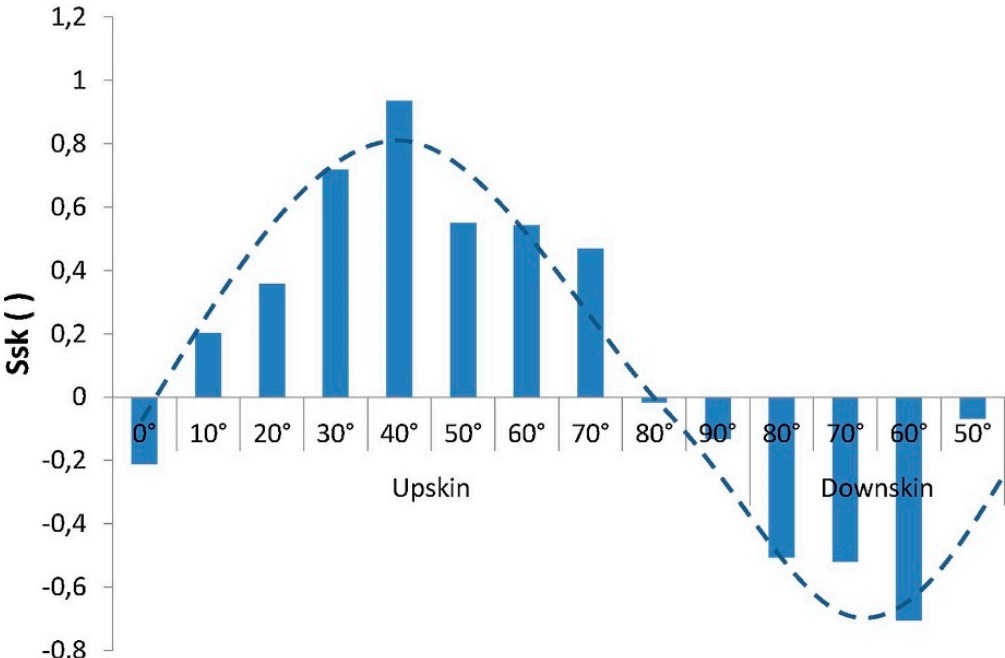

**Figure 1.** The relationship of downskin and upskin angles to skewness for SLM Ti-6Al-4V with the permission of Copy Clearance Center [55].

Other surface roughness parameters such as $S_p$ (line highest peak), and $S_v$ (maximum peak difference below average surface height) were also studied in the AM literature. Eidt et al. [57] studied these general surface roughness changes through adjusting the downward-angle tilt of laser powder bed fusion (LSBF) for the fatigue performance of Inconel alloy 718. In their study, two different specimens were manufactured from the LSBF process. In one specimen, the cube geometry consisted of four 90° built surfaces, whereas the second specimen consisted of a specialized cube that contains one 60° decline surface. The cubes were all measured as 0.5 inches in both the vertical and horizontal directions.

For the 90° angle-based parts, the power (P) acted as a dependent variable changing from 80, 100, and 120 W, whereas the speed (S) was held constant at 560 mm/s. P for the downskin sample was 75, 95, and 115 Watts at a speed of 785 mm/s. In the case of the 95-W power, the speed was 715, 750, and 785 W. Understanding that there is a lack of solidified material on down-skinned surfaces, the heat transfer of the heated particles will indeed have a different effect compared to vertical surfaces. It was found that for the 90° surfaces, $S_a$, $S_p$, and $S_v$ were all shown to decrease as power increases. Given this, it can be concluded that the complete melting of these particles resulted in a more even finish, which, in turn, improves the fatigue performance of the surface as there are fewer micro-cracks and micro-voids on the material surface. That said, similar results were also noted for the downskin samples as an increase in laser power reduces the maximum valley depth of the surface. With regard to laser speed, lower scanning speeds resulted in a refinement of surface parameters $S_a$, $S_v$, $S_{sk}$. As the speed increased, the powders most likely were stuck to the melted surface, thus increasing the maximum peak height. With higher peaks, there will be an increased chance that the tips of the asperities will break and wear on the surface, which can induce a greater wear rate in contact-based energy application.

However, despite these findings, the most commonly used parameter to represent the surface roughness is the average roughness, $R_a$. Although the other listed variables are categorically listed as amplitude surface parameters, surface roughness by itself has

been highly studied due to its direct impacts on the surface quality of AM-based materials. Kaji and Barari [58] are among many who have noted this trend. In their publication, layer thickness from fused deposition modeling (FDM) was found to have a correlation with surface roughness, as thicker layers resulted in higher surface angles which, in turn, negatively impact surface roughness. Through numerical modeling and experimental work, as the spacing difference between the plates increased, the influence of heat accumulation decreased, which allowed for a more optimized thermal gradient. Because of this, surface roughness was shown to decrease. With the discussion of surface angles, thermal gradients, and surface roughness, Strano et al. [59] were able to find a relationship between these aspects in SLM 316L alloy parts. Specifically, as the sloping angle increases, the part surface roughness took a staircase-like effect, decreasing in roughness as the sloping angle increased. The authors attribute this to the thermal gradient being optimized which, in turn, minimizes the effect of Marangoni convection.

The relationship of machine performance in terms of friction, wear, fatigue, and various other parameters has been correlated with surface roughness using $R_a$ as previously discussed. Through these relationships, the potential of increased durability, reduced maintenance periods, conservation of material, increased life of the equipment, and increased reliability has been achieved in conventionally manufactured components as the average surface finish is refined. Sheikh and Mishra support this statement in their review of tribology and its impacts on various Indian industries [60]. By analyzing the average energy consumption from surface properties such as friction and wear, they were able to determine the impact of tribology in both the automobile and manufacturing industries. Based on these calculations, a total of 55% of energy in manufacturing industries alone is lost to friction and wear. Considering that many of these industries are marked in the billions, this acts as a reminder of how crucial these parameters are. In AM-laser-based processes, the surface roughness of the component primarily depends on the laser processing parameters, which are used to calculate a single value called laser energy density. Laser energy density is defined as the laser power ($P$) divided by the scanning speed ($v$), thickness ($t$) and the scan spacing ($d$), or more commonly seen as

$$E = \frac{P}{v \cdot t \cdot d} \left( \frac{J}{mm^3} \right) \qquad (1)$$

A vast amount of the literature alludes to this key relationship as well. In Figures 2–4, summarizations of different experimental studies relating energy density to surface roughness are given, which are indicative of surface quality with regard to the particle. Although surface roughness is one of the more investigated surface variables found in the literature, it should be mentioned that surface roughness parameters such as surface waviness ($W_t$) have also been studied as a function of surface quality. One example can be seen in Gharbi et al., where surface finish induced by direct metal deposition (DMD) was studied for Ti-6Al-4V alloys [61]. Based on their results, $W_t$ was shown to have a direct impact on the melt pool quality. More specifically, as $W_t$ decreased, a smaller meniscus was created, which was indicative of deeper melt pools for the material-enhancing particle fusion.

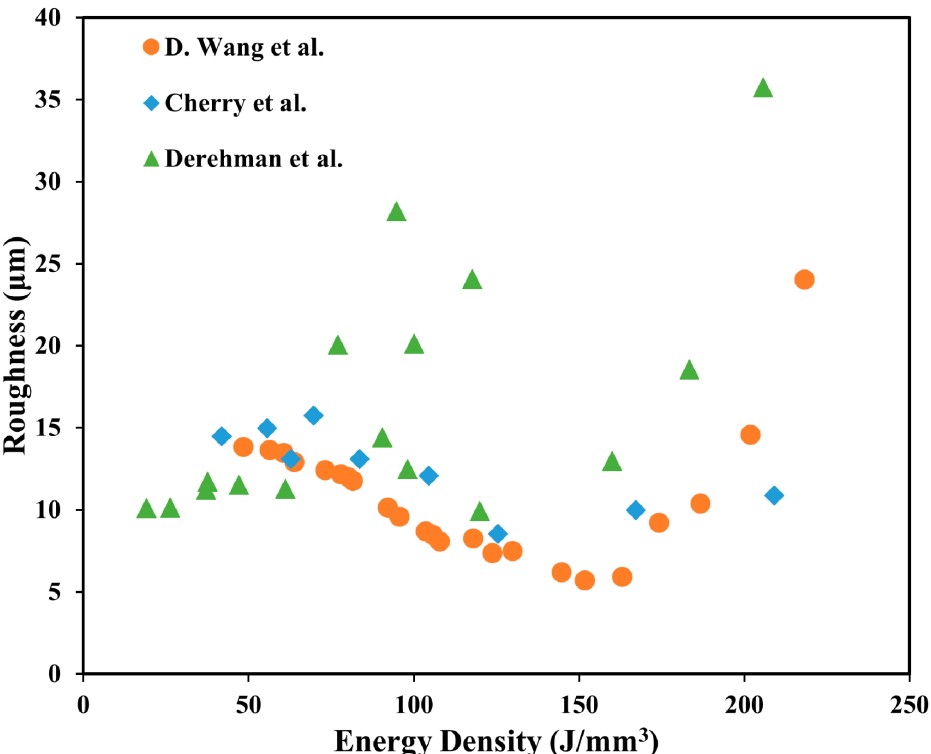

**Figure 2.** The relationship between energy density and average roughness ($R_a$) for P-LBF 316L steel.

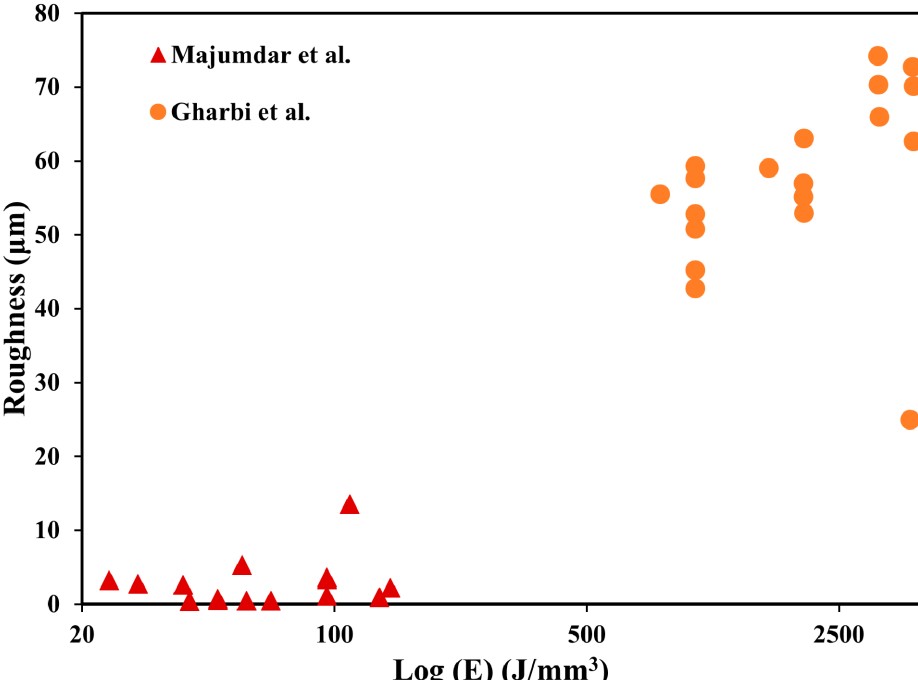

**Figure 3.** The relationship between energy density and average roughness ($R_a$) for Ti-6Al-4V in DMD systems.

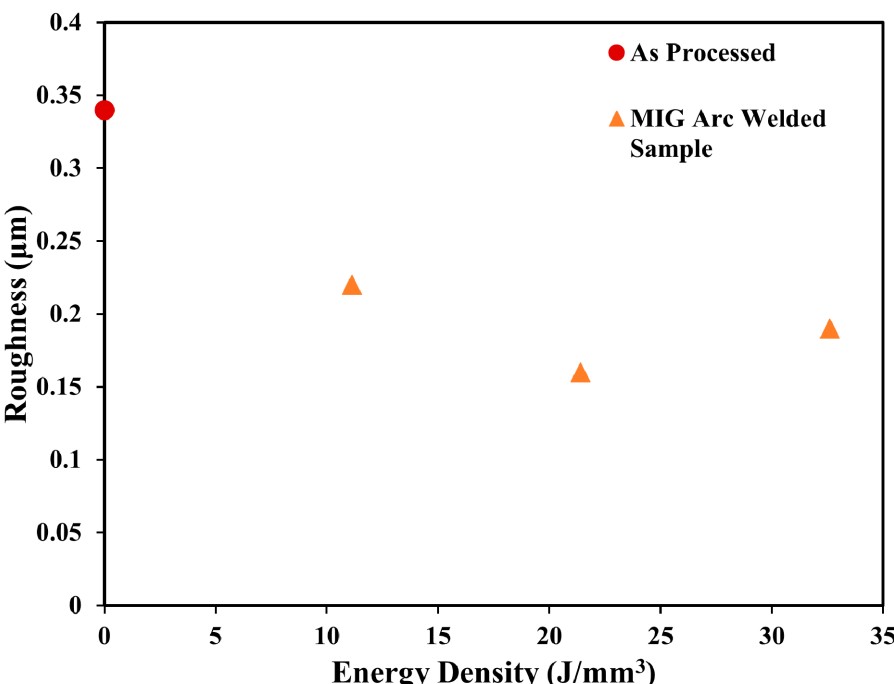

**Figure 4.** The relationship between energy density and average roughness ($R_a$) for an MIG arc welded ER5356 aluminum alloy.

In the case of Figure 2, the listed values indicate the relationship between the surface roughness and energy density of 316L steel melted from L-BFD. In order to explain this significance, each study will be individually discussed, which will allow for a greater appreciation and understanding of the trends found in the listed figure. In the case of the work from Wang et al, the laser energy density analysis can be broken into three different segments [62]. Below the 75 J/mm$^3$ region, the temperature from the laser was not high enough to induce a proper melt, which resulted in some sintered form from the particles. As energy density gradually increases (75–120 J/mm$^3$), fewer defects occur, and the lack of wettability due to insufficient melting results in a balling effect. This balling effect is due to the heated particles shrinking, thus taking a ball-like form, which can negatively impact the tribological and mechanical performance of the material. From 120–170 J/mm$^3$, the metal powder achieves an optimal amount of heat input to fuse with each other, thus increasing the wettability and smoothness of the track. Anything beyond the 170 J/mm$^3$ will result in overheating. As the particles begin to overheat, the melt pool will vaporize and often splash, with consequent defects in the aspects of material loss and roughness. Similar results were also found in the work of Cherry et al., as the laser energy density was shown to have a direct impact on both porosity and the general hardness of the 316L steel [63]. Specifically, within the 100–150 J/mm$^3$ region, the hardness was shown to be at its peak. The authors attribute this to the optimized melting of material decreasing its porosity and densifying the material surface. Derahman et al. also contribute evidence of this trend computationally and experimentally in their work [64]. At lower laser densities, the self-balling effect occurs on the surface. However, in the 100–150 J/mm$^3$ range, there seems to be an optimization of the surface finish, which in turn promotes the solidification of the various particles present in the build, which can largely impact mechanical and tribological performance. It should be noted that these studies only apply for 316L steel. Although these values may not be universal, the same processing mechanisms apply to all metal-based materials depending on their thermal conductivity and melting points.

In Figure 3, the relationship of surface roughness with energy density for DMD Ti-6Al-4V is shown. In the case of study from Majumbar et al. the laser energy density was varied from 20 to 140 J/mm$^3$ [65]. From 20–35 J/mm$^3$, there is insufficient melting (i.e., smaller melt pool size) on the working surface. As discussed for P-LBF 316L steel, the incomplete

fusion of particles results in a poorer surface finish. At these operating ranges, the lack of dynamic equilibrium of the melt pool results in a Plateau–Rayleigh instability in the process. As a consequence, the total surface area of the droplets decreases, despite the fluid volume being the same. With a lack of surface area, there is a greater chance of pores existing on the worked surface. In the 40–100 J/mm$^3$ range, there is a sufficient rate of power, resulting in a finer surface finish. In the 100–140 J/mm$^3$ range, there is an instability of the melt pool, with the melted particles taking on a Marangoni effect. During this effect, surface tension increases, creating a temperature-dependent gradient along with the melt pool. The melted pool is then subjected to a thermo-capillary convection flow, which in turn increases surface roughness. Continuing past this point, Gharbi et al. further contribute to this trend by analyzing surface roughness finish to laser density from 800–1400 J/mm$^3$ [61]. As expected, the roughness shows a similar increasing trend as the laser energy density is increased. In addition to this, the authors also came to a similar conclusion, as there was an increasing presence of the Marangoni effect as the melt temperature increased. Understanding this, the melt pool may not have enough cooling time, which can cause an improper fusion of the particles, thus increasing surface roughness. Based on this information, to optimize surface roughness, energy density should ideally be set within the 40–100 J/mm$^3$ range.

For MIG arc-welded ER5356 wire-based aluminum alloys, Zhang et al. also show a similar relationship, which can be explained through the same melting mechanisms, as shown in Figure 4 [66]. In the case of this alloy, after the 30 J/mm$^3$ range, a stair-stepping effect occurs due to molten flow from the increased heating. Given a proper optimization of energy laser density (between 20 and 25 J/mm$^3$), there is a better fusion of the aluminum particles, resulting in a finer surface roughness and finish.

Post-processing techniques have been used in the literature to mitigate some of these defects, however, the optimization of processing variables is highly needed in order to further the progression of AM technologies.

### 1.5. Current Challenges in Additive Manufacturing

Through the significant proliferation of novel additive manufacturing technologies, there are many challenges that have arisen for metal-based parts. In general, there are three core issues that are present with 3D printing-based technologies. The first general issue that is seen in additive manufacturing technologies is the general high cost of different technologies. Some technologies are retailed in the hundreds of thousands, whereas others can be in the millions (USD). One example of this can be seen in Thomas and Gilberts' literature review of the general costs of additive manufacturing technologies [67]. Through a general cost analysis of several industrial firms, costs regarding machine costs, material costs, energy consumption, and general labor were all factored and elaborated on. As a general takeaway, AM technologies tend to be quite costly, especially newer models, due to their limited production quantities and availability in the market. Khajavi et al. are also among the researchers who have noted the cost impacts of AM technologies in industrial supply chains [68]. In this specific study, the impacts of AM on the aviation industry for F-18 Super Hornets were analyzed. Taking into consideration the amount of time, cost, and labor required to use an AM, the highest cost per annum for this instance was marked at USD 1,793,971, which is incredibly costly considering the general slow production of AM technologies vs. traditional manufacturing technologies. Although industries such as healthcare and medical benefit long-term from the flexibility of AM technologies, other larger production volume-based industries (e.g., automotive) maybe favored more traditional manufacturing processes due to the lack of mass production. It should also be mentioned that the lack of versatility for fabricating larger structures is an issue with AM processes, as well as the processing speeds and poor dimensional accuracy of these processes.

The second and third issues in AM are the anisotropic mechanical properties of the build direction and the defects and limited materials that can be used for these processes. In general, one of the primary issues seen in AM components is the formation of voids and

pores, which Sola and Nouri [69] stated in their literature review of the microstructural porosity of PBF-based metals. Sola and Nouri state that this can be largely attributed to the thermodynamic conditions of the melting process, as the processing parameters dictate the finish of the part. In the case of a larger press, the mechanical performance of metal-based parts drastically reduces, potentially causing structural failures in critical operations.

With the many challenges of AM components, many novel processes have been added to the fabrication process of AM metals. The surface defects of the melting process tend to be the primary issue due to a *Stair Casing Effect* during fabrication, which in turn deteriorates the surface roughness of fabricated components. An example of the *Stair Casing Effect* is shown in Figure 5.

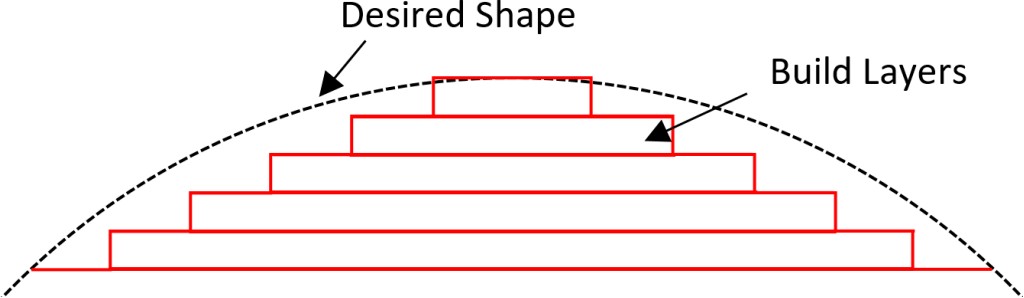

**Figure 5.** A schematic of the Stair Casing Effect that has been shown in AM processes.

The stair casing effect, in its most basic definition, is the curvature limitation of layer-building processes, as the concavity and convexity of the CAD-generated radius tends not to be as easily replicated in actual production. Yasa et al. are along many who have come to this conclusion for AM-based metals [70]. Through the use of computer-based simulations and experimental validations, the waviness of the two consecutive layers was studied and compared, as the authors used this variable as a key indicator of the stair casing effect due to its resemblance to stair casing. Aside from the utilization of different processing variables, post-processing techniques are commonly used to mitigate these issues. In Kumbhar and Mulay's literature review of the surface-finishing technologies used in AM, they identified post-processing methods, to be segmented into two different categories: conventional and non-conventional [71]. For conventional methods, those most commonly seen in scientific studies are as followed: (1) micro-machining processes (MMP), (2) CNC finishing, (3) hot cutter machining (HCM), (4) vibratory bowl abrasion, (5) optical polishing, (6) epoxy resin painting, (7) friction stir processing/welding, (8) laser re-melting, (9) shot peening, (10) heat treatment. Non-common methods are also as follows: (1) laser-micro-machining, (2) electrochemical polishing, (3) ultrasonic abrasion, (4) electroplating, and (5) chemical post-processing treatment. The authors of this paper were able to support these claims through referencing various publications regarding this topic.

Aside from the listed post-processing techniques, an in situ technique known as hybrid AM is another avenue that can be used to improve the integrity of manufactured parts. This concept was highlighted by Merklien et al., where the current state of hybrid AM technologies were reviewed and detailed [72]. Based on this review, hybrid AM is defined as the application of two or more manufacturing processes used to improve the part quality of AM components. Generally, these techniques are done through hybrid machining, hybrid materials, and hybrid processes, which can help aid the final finish of AM energy-consuming products.

In one sense, through synergistically combining different additive and subtractive processes, the quality of manufactured products drastically increases. Flynn et al. have identified that through these specialized and sequential-based systems, parts can achieve more complex geometrical shapes and limit any issues regarding overhanging and internal characteristics [73]. The authors support this claim through their combined analysis of commercially available machines combined with the most up to-date literature regarding

additive and subtractive processes for hybrid AM. These combined processes are especially helpful in low-volume, high-cost operations where equipment such as in the biomedical and weaponry field allow for a reduction in lead timing and specialized tool costs that would be normally found in traditional manufacturing processes. However, one crucial downfall of both additive/subtractive and post-manufacturing processes is the cost and time required for these processes. With many publications in this topic being published (including patents) within the last 5 years, AM process optimization has yet to be fully investigated. Given the improvement in process enhancements, there will be less of a reliance on the additional post/in-situ processes, which will help improve the total quality and performance of AM parts. This paper aims to collectively display all of the current literature on how to maximize surface quality from processing variables and how this can improve other surface properties such as hardness or wear. This will allow for less dependence on additional post-manufacturing processes and improvements in the general qualities of the AM part.

## 2. Additive Manufacturing Techniques

Three primary processes of AM techniques have been adopted in the manufacturing of metal and alloys as components. These are: (i) powder bed fusion (PBF), (ii) powder feed systems (PFS), and (iii) wire fed systems (WFS). Although these processes use similar techniques, small deviations in the tooling style allow unique fabrications to have various advantages and disadvantages in the fabrication of different metal systems. Table 3 lists the metal-based systems and differentiates the respective preferred AM techniques for manufacturing.

**Table 3.** A summation of the common processes and materials associated with PBF, PFS and WFS.

| Classification | Associated Processes | Commonly Used Metallic Materials | References |
|---|---|---|---|
| **Powder Bed Fusion** | Direct Metal Laser Sintering (DMLS) | Steel, Ti-6Al-4V, AlSiMg, AlSi10Mg, SiC | [74–80] |
| | Electron Beam Melting (EBM) | H13 Steel, 316, 316L, AISI 420, AISI 4340, cp-Fe | [74,75,80] |
| | Selective Laser Melting (SLM) | AlSi10Mg, cp-Al, TiC, CoCrMo, Co-Cr, 316 Stainless Steel, Inconel 718 | [74,75,80–86] |
| | Selective Laser Sintering (SLS) | Fe, Ni, Cu, Fe₃P, WC-Co | [74,80,87] |
| **Powder Feed Systems** | Laser Metal Fusion (LMF) | Fe, Al-Alloys, Ni, Co-Cr | [80,88,89] |
| | Laser Engineering Net Shape (LENS) | Tic/Ti, 316 Stainless Steel, Ni-Alloys, Ti, H13 | [75,90,91] |
| | Laser Metal Deposition Powder (LMD-p) | Cu, 316 Stainless Steel, Inconel 690, Ti-6Al-4V, TiC | [80,92–94] |
| | Direct Metal Deposition (DMD) | AISI 316L Stainless Steel, H13, Inconel 625 | [75,80,95–98] |
| **Wire Feed Systems** | Wire and Arc Additive Manufacturing (WAAM) | Ti-6Al-4V, Fe-Al, AZ31, Ni Alloys | [99–103] |
| | Electron Beam Freeform Fabrication (EBFFF) | 2219 Al, Ti-6Al-4V, 718 Alloy | [94,98–100] |
| | Laser Metal Deposition-wire (LMD-w) | Ti-6Al-4V, 316L, Alloy 718 | [99,103–108] |
| | Electron Beam Additive Manufacturing (EBAM) | Al Alloys, Tool Steel, Co Super Alloys, Ti-6Al-4V | [99,103,109] |
| | Shaped Metal Deposition (SMD) | Ti-6Al-4V, Ni-Alloys, Inconel 718, 300M Steel | [99,103,106,110–113] |
| | Additive Layer Manufacturing (ALM) | Ti-6Al-4V, HC101 Steel | [99,103,114,115] |

In the following sections, each classification of AM-based systems will be detailed and differentiated based on their working components and how the metals are applied to the system. This will allow for a basis on which to understand each individual system and their applications for 3-D manufacturing.

## 2.1. Laser Powder Bed Fusion

Laser powder bed fusion (L-PBF) is one of the widely used AM techniques to manufacture critical metal and alloy components. The other variations of L-PBF are direct metal laser sintering (DMLS), electron beam melting (EBM), selective laser melting (SLM), selective laser sintering (SLS), and laser metal fusion (LMF). L-PBF has been noted to be one of the fastest-growing AM techniques. L-PBF technique uses thermal energy via laser beams to melt the powder particles on the powder bed locally, and solidification of this yields a solid structure. Essentially, local melting and solidification processes are involved in the manufacturing of 3D metal and alloy components in L-PBF processes. Using metal powder as the raw material, various components can be fabricated for multiple engineering applications such as aerospace, healthcare, and automotive.

Figure 6 demonstrates the general PBF system, which consists of the following components: laser power, mirror scanner, roll, overflow, feed container, and build cylinder. The laser beam generated from the system is directed to the powder bed and deflected onto a sequence of scanning mirrors, which are controlled using a computer interface depending on the 3D design of components. During the laser exposure, the feed container deposits preheated material to the surface as a powder bed, where a re-coater (either in the form of a roller or a blade) helps to evenly distribute the material across the entire piston of the machine. With the laser radiating on the powders, any excess material is collected through overflow bins. The leftover powders are further reutilized for manufacturing with material waste in this process.

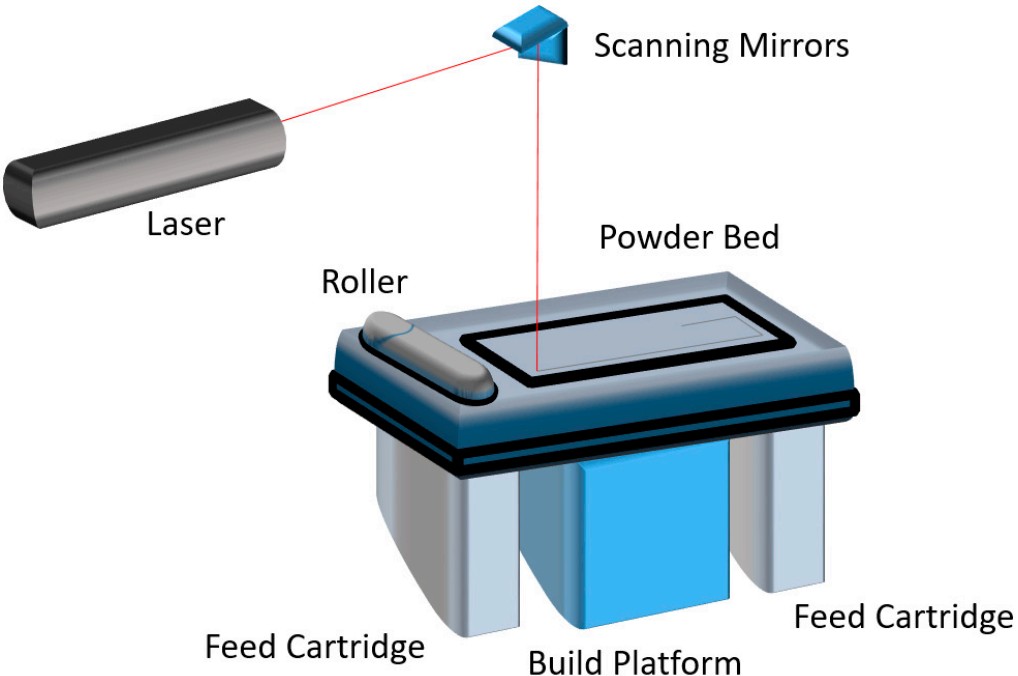

**Figure 6.** A schematic of a general PBF system.

The advantages of this technique include low-cost manufacturing, applied to a large number of metal and alloys, and easier composite manufacturing. What truly differentiates this method from many other methods is the extensive range of materials that can be used. Materials such as cermet, ceramics, metal–polymer powders, polymers and nylon, as well as metals and their alloys, can be applied, helping to develop extremely complex products.

## 2.2. Powder Feed Systems

Powder feed systems (PFS) operate through material deposition in conjunction with a focused laser beam. The difference between PFS and L-PBF is the methods of adding powders to the system. In PFS, material deposition is done through a designated nozzle

that deposits powder along with the location of the laser beam. As this nozzle deposits material, the given laser melts the material, allowing a melt pool to occur, thus allowing a proper disposition of material to the surface. These systems are composed of the following: (i) laser cladding, (ii) laser metal deposition, and (iii) directed energy deposition. One of the main advantages that this system has is the ability to build larger structures as well as the capability of repairing damaged components in comparison to other well-known AM technologies. A detailed figure of the general schematic of PFS is shown in Figure 7.

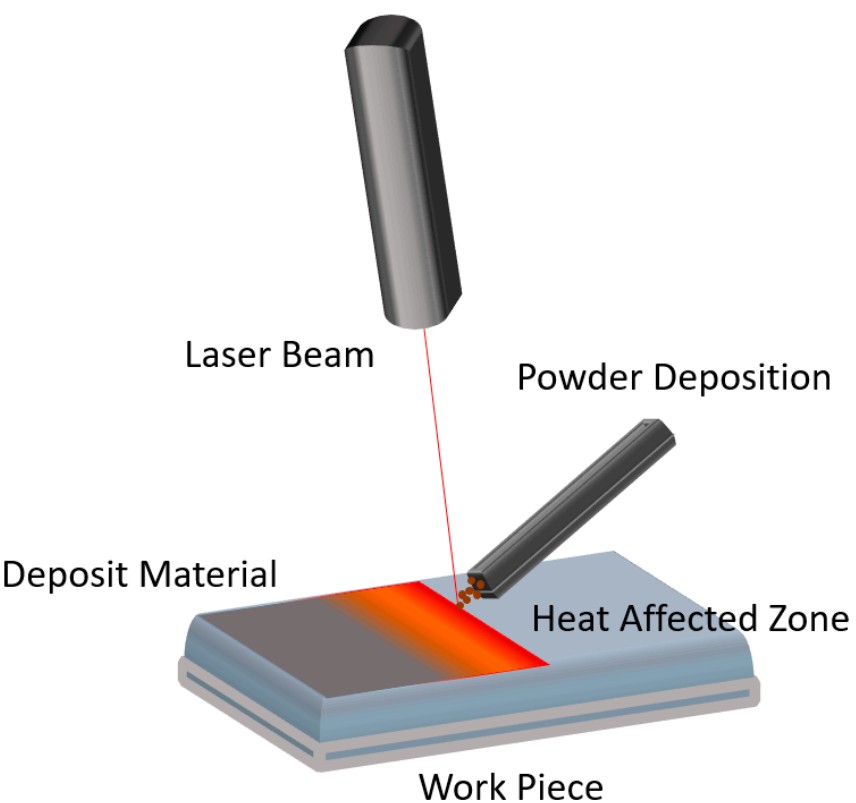

**Figure 7.** A general schematic of a powder feed system.

*2.3. Wire-Fed Systems*

In wire-fed systems (WFS), a wire of the component is used as the raw material for the process. As this impacts the surface, an electron beam melts the material, resulting in a similar layer building concept to the previous two AM techniques. However, what differentiates this technique from the others is its ability to manufacture large and complex metallic components. Typical materials that are used in WFS operations include titanium/titanium alloys, nickel, copper, stainless steel, aluminum alloys, and cobalt alloys, as listed in Table 3.

A visual schematic of how a WFS system operates is shown in Figure 8. A WFS generally contains an electron beam (EB) gun, electron beam, laser beam, laser wire feed, deposition layers, and the substrate itself. As this system operates, an automatic feedstock is utilized, which supplies the metallic wire to the substrate. As the wire is fed, the EB gun imparts an electron beam to melt the wire in a melt pool shape, thus solidifying the material and creating a layer-by-layer build. Through a pre-set program, the laser head can create larger build volumes of materials due to the high dispositioning rate in WFS.

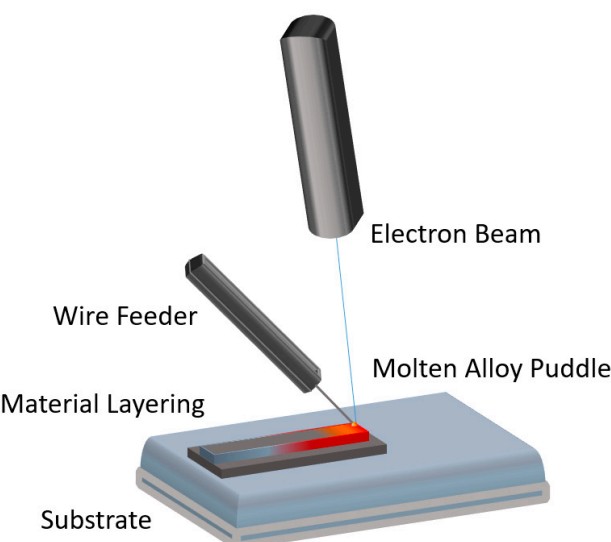

**Figure 8.** A general schematic of wire-fed system.

## 3. Materials and Tribology in AM

Currently, a limited number of metals is utilized in AM. Generally, these metals can be segmented into the following categories: (i) stainless steels, (ii) aluminum alloys, (iii) cobalt chrome alloys, (iv) titanium alloys and (v) nickel-based alloys based on the data from Table 3. Each of these different metallic groups tends to have different behaviors during AM-based operations, on top of the variability shown in different AM processes, their processing parameters, and the physical phenomena that occur as a by-product of this. The following sections will detail the literature regarding surface texture optimization and tribology for all of the listed metals, as well as listing the current challenges and future directions for these metals.

### 3.1. Stainless Steels

As one of the most commonly utilized materials in AM, stainless steels are held to an extremely high standard due to their cost efficiency and excellent mechanical properties. Seifi et al. are among many who have supported this statement in their review of metallic materials used in AM [116]. The authors support this claim through the amount of publications that are referenced using stainless, tool, and alloy-based steels. More importantly, due to their resistance to oxidation and corrosion, many trades, ranging from defense to chemical industries, utilize this material to optimize manufactured items. With common technologies such as selective laser melting (SLM) utilizing this material, intricate and efficient designs can be created, that otherwise would not be achievable with common manufacturing practices.

It should be mentioned before starting this section that there seems to be a lack of literature covering the aspects of mechanical properties, tribological properties and surface roughness finish for AM steels. Despite this void, this section will cover various studies that investigate combinations of these aspects and create links between whatever aspect is missing in the study. For steel-based AM, the literature review from Zadi-Maad et al. states that the main processing parameters of investigation consist of layer thickness, flow rate, powder mass, laser power, scanning speed, beam scan pattern, beam diameter, and build direction [75]. Zadi-Maad et al. further explore this by listing a wide array of studies including electron beam melting (EBM), SLM, direct metal laser sintering (DMLS), laser engineered net shaping (LENS), and direct metal deposition (DMD), which happen to investigate some combination of these parameters. Based on the previous sections, the main takeaway from these processing parameters is that through variations in temperature from the laser beams, material properties such as the grain size, grain orientation, and general surface finish of the material can all be significantly altered. One interesting point

to also take into consideration is the manipulation of spacing distance for laser melted steel. Jamshidinia and Kovacevic explore this topic in their experimental paper focusing on the impacts of heat accumulation on surface roughness finish [117]. Giving the spacing distance between the plates of 5, 10, and 20 mm, the authors discovered that spacing distance shares an inverse relationship with surface roughness parameters. Understanding that surface roughness can indicate how well particles are fused together, this is another important point to take into consideration when optimizing the performance of AM steel.

For maraging steel 1.2709, Monkava et al. [118] found a correlation between the base surface roughness and hardness for various orientations for an angled DMLS printed cubic part. In this study, the printed wall varied, with an angle range from 37° to 60°, with four specific testing areas. What was found was that the hardness values in the XY and YZ directions for the non-supported sample were superior to the supported side. For the XZ direction, the supported sample was comparable. In comparison to the average roughness, the lower the average roughness was, the higher the hardness was. One explanation for this was rooted in the thermal gradients that exist from the supports. Given a larger thermal gradient, inner stress may occur, which, in turn, can shrink the part, somewhat similar to the balling effect. With a larger amount of thermal stress being present on the supported side, there is a more likely chance that some regions may shrink more than others, thus causing an inconsistent surface finish and affecting the surface quality of the cube. Given that there is a lack of cooling on the working surface, the possibility of pores may occur, resulting in a less dense surface, which, in turn, can affect mechanical and tribological performance, as they are closely linked to each other. Given these improvements, there will be less maintenance on manufactured parts, which can reduce the energy utilized in manufacturing facilities.

Focusing more on tribological-based studies, Zhu et al. [48] are among the many researchers who have investigated the tribological properties of AM materials. In this study, SLM 316L stainless steel was investigated, where tribological performances were evaluated compared to traditionally processed (TP) samples. The SLM parameters that were used to fabricate the samples consisted of a 200 W fiber laser operating at a 1070 nm wavelength and a 70 μm beam diameter. The thickness of each layer was measured at 30 μm, with a scanning speed and point distance of 700 mm/s and 70 μm. With both SLM and TP samples polished to an $R_a$ value of 0.8 and 0.02 μm, the friction and wear values of the SLM samples were measured to be slightly lower than TP samples when in contact with brass. However, when in contact with a harder material (38CrMoAl), the changes in friction and wear values were much greater. Based on a topographic and microstructural analysis, this was attributed to the refined grains in the SLM sample, reducing the quantity of pores present. If the quantity of pores can be further reduced via increased material densification, it can be predicted that the tribological performance of the sample can increase in comparison to TP substrates. If there is a generally lower value of average surface roughness, this will indicate fewer pores along the surface, which improves the densification of the material.

Shibata et al. [119] are among the other researchers who have studied the effects of lubricated wear conditions on AM components. In their study, JIS SUS316L powder-based steel was fabricated by PBF under five different processing conditions and tested under oil lubrication conditions. The conditions for the PBF-fabricated samples are shown in Table 4.

**Table 4.** The PBF conditions used for the samples in the Shibata et al. study [114].

| Fabricated Samples | Laser Power (W) | Scanning Speeds (m/s) |
|:---:|:---:|:---:|
| Sample 1 | 240 | 0.7 |
| Sample 2 | 280 | 0.4 |
| Sample 3 | 280 | 0.7 |
| Sample 4 | 280 | 1.0 |
| Sample 5 | 320 | 0.7 |

Given these operation conditions, the energy density was ultimately calculated from these conditions and used as a reference for determining the tribological performance of the samples. In the same order as the listed samples in Table 4, the laser energy densities are as follows: (1) 45.7, (2) 93.3, (3) 53.3, (4) 37.3, and (5) 61.0 GJ/m$^3$. Given that the testing surfaces were polished to an average surface roughness (Ra) of 0.001 μm, a high-carbon chromium-bearing steel ball was used for the reciprocating wear tests at normal loads of 1.96, 4.90, and 9.81 N. It was observed that all the samples yielded a large COF value during the initial parts of the wear test for all loads. Over time, the COF values began to stabilize. The takeaway from these results is that the asperities from the initial stages of testing are at their maximum roughness. However, due to the breakage and deformation of the peak asperities, they begin to flatten, thus reducing the COF and optimizing the desired results. In this study, sample 5 was shown to have the lowest mean friction coefficient and wear rate out of all the tested samples. The results can be explained through the hardness (3.198 GPa) and porosity (1.42%) values of the sample, which were quite intermediate compared to the other four samples. This would indicate that if the hardness is increased, surface defects are reduced, which could be detrimental, as oil pockets can form during testing. Likewise, if the hardness is any lower, then the surface will become less resistant to any loading effects, thus resulting in an increase in friction and wear.

Given the current state of the literature, the authors suggest that additional research should be conducted on the relation between surface roughness finish and the tribological/mechanical properties of AM laser-based steel. Based on the discussed studies, there seems to be a clear relationship between these parameters, which can be quite helpful when minimizing the amount of post-processing machines needed to optimize the performance of these materials.

### 3.2. Aluminum Alloys

Similar to stainless steel, there is a limited number of published articles describing the tribological effects of additive manufactured aluminum alloys. With selective laser melting/direct metal laser sintering (SLM/DMLS) being the main processes that utilize this material, Manfredi et al. identified in their review of AM Al Alloys that the primary types of aluminum used are A6061, AlSi12, AlSi12Mg and AlSi10Mg [120]. Manfredi et al. came to these conclusions through the thorough analysis of various sources in the literature, as they compared these to various material properties of these alloys post-fabrication. Through the utilization of aluminum alloys, their high strength properties and corrosive resistances proved to be greatly useful in industries such as in aerospace, where lightweight materials tend to be more preferred compared to other, heavier metals. In addition to the modification of mechanical properties, AM techniques can additionally improve other properties, such as the microstructure of the material, potentially allowing for a higher degree of structural integrity and anisotropic mechanical performance. Rosenthal et al. [121] is among many who have found an improvement in anisotropic mechanical performance for SLM AlSi10Mg. Through the impacts of partial re-melting, the porosity of the investigated samples decreased, which, in turn, allowed for greater particle bonding, thus allowing a denser, compact, and fine finish. These improvements were then validated through evaluating the elongation properties of the substrate, as they improved as a byproduct of this.

However, the lack of available and reliable powders available for the feedstock is a limitation for aluminum. Given the fundamental differences between traditional and additive manufacturing, factors such as powder weldability and solidification rates are significantly different, resulting in unwanted voids throughout the build. Aboulkhair et al. elaborate on this fact in their review of the impacts of SLM for AM Al alloys [122]. Particularly, the authors identified that through the impact of laser irradiation, the solidification of the metal depends on various factors, such as the laser intensity, the scanning speed, and the general type of aluminum used. Because of this, factors such as the melt pool evaporation, as well as the relating heat transfer and chemical reactions which cause the formation of

the build, take place, which can directly relate to both the surface finish and the materials' properties. With aluminum being an easily meltable material, many surface defects such as voids may occur due to unoptimized melting conditions. One example of this can be seen in the publication by Kempen et al. [123], where AlSi10Mg was processed via SLM. It was found that in order to optimize the alloy's performance, a series of parameters must be fulfilled during testing. These parameters consist of having an uninterrupted scanning track, having the laser penetrate the previous layer, and having an appropriate height and a connection angle of 90°. However, regardless of this, continuous advancements in improving the processing of this metal have taken place throughout the past decade, negating some of the previously stated cons.

It should also be taken into consideration that other surface defects, such as balling effects and surface satellites, can occur. Aboulkhair et al. further detail this in their review, suggesting that these surface defects tend to be one of the core issues associated with the AM of aluminum alloys [122]. In essence, the authors describe these balling's effects as the byproduct of the partial melting of particles throughout the laser process. Because of this, a ball like feature appears on the surface which, in turn, can negatively impact the deposition of new layers, resulting in unwanted porosity. Relating this to surface roughness, balling may indicate the quantity of surface defects present on an as-finished build, which, in turn, can directly impact the mechanical and tribological properties of Al Alloys.

Mower and Long are among many who have encountered these defects from the AM of aluminum alloys [124]. In their study, surface defects in the form of voids were found in SLM'd AlSi10Mg. In fatigue testing, these voids were shown to have a greater likelihood of crack propagation from the continual stress-induced overtime. This was especially true in regions where clusters of micro-voids were present. However, it should be noted that optimized cooling rates may contribute to the mitigation of surface defects, as well as improve the quality of the surface which, in turn, improves the fatigue properties of the alloy. Despite this, research on this topic is extremely limited and has yet to be fully explored. This is especially apparent with the evaluation of non-mechanical properties such as corrosive resistance and lubrication manipulation. This topic of research would be extremely beneficial to explore for any contact-related applications, as less energy will be consumed, which can reduce manufacturing costs.

### 3.3. Cobalt Chrome Alloys

Unlike the other metals mentioned in this list, cobalt chrome alloys are often viewed as problematic due to the variation in mechanical anisotropy with regard to their build orientations. As a consequence of this, the performance of the metal can suffer throughout various applications. Hitzler et al. [125] are among many who have encountered this in their study focusing on the impact build inclination of Co–Cr–Mo- and Co–Cr–Tungsten-based dental alloys. One of the key things that was noted was that the cooling rates varied with each build orientation, which in turn affected their mechanical anisotropic properties.

Within the literature, a vast majority of additive manufactured cobalt chrome alloys tend to be centered around dental applications. This can be attributed to the impressive mechanical properties they exhibit in partial denture applications, as well as their excellent resistance to corrosion. Through multiple publications there appears to be a trend of SLM technology, increasing the various properties of these metals, which could greatly be beneficial to such dental applications, as well as various other applications. On example of this can be shown in the publication from Wu et al. [83]. In their study, the mechanical performance of Co-Cr alloys fabricated via SLM was investigated and compared to the cast alloy that is usually used in dental applications. What the authors found was that the SLM alloy exhibited far superior mechanical strength in the form of yield strength and tensile strength. The explanation for these findings can be attributed to the dense and compact finish of the SLM process. Generally, with metal ceramics, the porcelain bond strength dictates the performance of the substrate, as it helps the metal with the resisting repeating mastication forces that occur over time.

As of now, there are no studies that focus on the processing optimization of AM cobalt-based materials with specific mention of surface roughness and its relation to tribological and mechanical-based properties. The closest form of research that touches on this topic can be found in a study by Ţălu et al., looking at post-surface-finishing techniques in the forms of electro- and mechanical-based polishing [126]. However, the aim of this study was only to focus on the final surface finish, rather than mechanical- or tribological-based performances. Sidambe closely touches on this subject as well, in the form of evaluating the surface topography of as-finished SLM cobalt–chromium–molybdenum (Co–Cr–Mo) as a function of build orientation. However, this study only focuses on the proliferation of L929 mouse fibroblast cells, rather than on mechanical and tribological performance. Keeping these studies in mind, there is an opportunity for a new avenue of research to be explored.

### 3.4. Titanium Alloys

Being one of the most investigated materials in general manufacturing, titanium alloys are held to extremely high standards in various industries, ranging from biomedical to aerospace. This can be mainly attributed to the high strength-to-weight ratio that this material possesses. However, within AM, the use of titanium is not as extensive, due to its high costs. Similar to the other metals mentioned, the microstructure and mechanical properties are primarily investigated with titanium alloys when subjected to AM techniques. Baurfeld et al. test the impacts of shaped metal deposition (SMD) by building various different geometries from Ti-6Al-4V, where surface quality and mechanical properties were evaluated [110]. From the top region of the components, it was found that with a greater surface finish, the microstructure and ultimate tensile strength (UTS) were greatly improved from the phase change induced from processing. When assessing these changes (especially in microstructure), the impact of cooling rates, as well as the materials' thermal history, can drastically influence these properties. In the literature review from Debroy et al., the series of studies that was referenced eluded to temperature gradients from melt pools having a significant impact on the phase transformation of the worked alloys, which can affect a variety of the mechanical properties of the build as well as fatigue-based performance in relation to surface roughness [127]. Understanding this, the vast majority of studies, including this alloy, are primarily centered around Ti-6Al-4V due to its superior properties in comparison to other existing alloys.

With regard to exploring the relationship of surface roughness with mechanical fatigue performance, Edwards et al. [128] investigated this by studying the build orientations of an SLM Ti-6Al-4V substrate. In this study, machine settings were as follows: scanning speed (200 m/s), laser power (200 W), laser wavelength (1070 nm), point distance (50 μm), point exposure time (251 μs), and energy density ($11.1 \times 10^{-5}$ J/mm$^3$). Build orientations were set in three orientations being the x, y, and z. Results in this study indicate that the z-direction yielded the highest surface roughness ($R_a$) value, measuring at 38.5 μm, compared to the x and y directions measuring at 32.0 and 30.3 μm. Microstructural and fatigue tests also indicated a proportional relationship to these values with regard to the negative fatigue performance of the material. Being tested at 200 k cycles to failure, the x-direction force measured 240 MPa, whereas the y and z directions measured at 170 and 100 MPa. In addition to this, the z-direction indicated that it was aligned with the identified columnar morphology, demonstrating that as a material is subjected to fatigue testing, the orientation of the material with respect to its build will have a significant impact on the total values. This is also supported by the work of Thijs et al., where build orientation had a direct impact on the grain structure and surface roughness for SLM'd Ti-6Al-4V. As the scanning speeds and build directions varied, different impacts on the phase transformation and surface roughness occurred. When the melt pool had an enhanced cooling rate, a martensitic phase transformation occurred, which helped improve the mechanical hardness of the material. If the melt pool was given any less time to cool, surface defects would occur, thus creating a courser surface finish than what would have been found otherwise.

In relation to surface quality and fatigue, Wycisk et al. similarly found a direct relationship between these qualities for SLM'd Ti-6Al-4V [129]. As surface roughness was improved, there was less of a tendency for voids to be present on the material. With an increase in voids, there is a greater chance of crack initiation which, in turn, would cause an early failure of the part, which overlaps with the previous studies discussed in this review. In a separate study from Mahamood and Akinlabi, the surface-finishing effects of Ti6Al4V were investigated, where variations in the powder flow rate acted as a catalyst for improved surface roughness [130]. With the processing parameters consisting of a stead laser pulse rate of 4.0 kW, as well as scanning speed and gas flow rates of 3.0 kW and 0.004 m/s, a total of six experiments were conducted. With the powder and substrate material consisting of Ti6Al4V, Mahamood et al. found that the powder rate shared a directly proportional relationship with average surface roughness, with the lowest recorded values being a powder flow rate of 1.44 g/min and surface roughness of 4.5 μm. This can be attributed to the size and efficiency of the melting pool generated throughout the process. With an increase in the powder flow rate, the material experiences an improper melting rate, thus solidifying the material at a faster rate, resulting in the poor surface finish. As flow rate decreases, the melt pool size increases, allowing for the powder to better merge and solidify with the substrate. This is also supported in the other literature sources, such as those from Gharbi et al. [61]. In this publication, DMD Ti-6Al-4V was studied, along with the laser power, laser velocity, and mass feed rate of the material powder. In unoptimized conditions, surface degradation acted as a function of the sticking of non-melted particles and the buildup of menisci. Because of this, it can be insinuated that with an improvement in surface roughness, the surface properties of the substrate will be enhanced, as shown in the other discussed studies of this review.

*3.5. Magnesium Alloys*

Magnesium (Mg)-based alloys are also another sought-out material that are frequently used in weight-sensitive operations. Compared to other commonly used metals, the density of Mg ranks quite low due to its low atomic mass. In biomedical and automotive, and especially aviation and aerospace industries, Mg allows for scientists to create a castable, strong, but lightweight part that excels in performance when compared to metals such as aluminum. Jahangir et al. elaborate on this point by giving specific examples of the advantages of Mg over other metals [131]. One of the examples that was provided pertained to the use of bio-implants in the biomedical industry. Specifically, Mg has preferable degradation properties that tend to be better-fitting for implants. The various mechanical properties are also shown to be closest to natural bones which, in turn, helps to improve the osseointegration process of the surrounding cells. Relating this to the concept of tribology and surface roughness, various bio-implants tend to be subjected to wear-based conditions from the surrounding bones. In addition to this, surface roughness can also indicate the likelihood of osseointegration with the surrounding tissue. If AM can be optimized to enhance both of these properties, it is a given that there will be a greater chance of success for implant-based operations, that may help to preserve the bioimplant for a longer duration of time than it would have otherwise.

In aviation- and aerospace-based industries, Withers and Mishra note that magnesium is also critical, as tens of millions of gallons of gasoline are saved as a byproduct of the weight reduction for widely used aircrafts [132]. The authors detail that due to its lightweight nature, complex geometrical parts can be easily designed and used, which can greatly help with aircraft performance. However, despite these advantages, Karunakaran et al. state that one core issue with the AM of magnesium is the oxidizing effects that magnesium has when exposed to oxygen [133]. Especially in an environment that utilizes lasers (i.e., AM), if the laser intensity is not controlled, it could be highly reactive, and could potentially have a risk of combusting under un-optimized conditions. Karunakaran et al. explain the implications. Nonetheless, with additional research, these effects can be minimized and mitigated.

To date, a large percentage of the AM literature for magnesium alloys involves the usage of laser-based technologies. In most cases, SLM is the most commonly used technology. One exemplary review that fits this criterion can be found from Kurzynowski et al. [134]. In this review, the authors direct their focus to the recent advancements of SLM for Mg alloys, specifically in the aerospace industry. Tying this back to the fact that aerospace industries often use Mg alloys due to their molecular weight and mechanical performance, there has been a drastic increase in its usage in conjunction with AM-based technologies. With regard to surface finish optimization and the mechanical and tribological performance, optimization of the surface finish can be indicative of the overall quality of the build as well as an improvement in performance, as the particles are more greatly bonded together. Savalani and Pizarro contribute to this claim in their paper focusing on the effects of pre-heating of Mg particles prior to SLM [135]. In this study, the as-finished product of surface roughness in conjunction with build thickness was studied in conjunction with mechanical performance. The results from the study indicate that the finished surface roughness shares a relationship with layer thickness; as layer thickness increases, there is a larger number of voids and pores present on the surface from Marangoni convection. For a layer thickness of 0.15–0.20 mm, the hardness and elastic modulus of Mg was greater than for a layer thickness of 0.25–0.30 mm. Despite no tribological testing being present in this study, it can be safe to assume that through the improvement in the mechanical properties as well as the density of the deposited layers, there could also be an increase in tribological properties.

Similar to the contents from Section 1.3, laser energy density is one factor that highly influences the overall part quality of AM magnesium. Generally, if the laser energy density is too high, the heightened temperature will increase the vaporization of the interjected magnesium particles. With an enlarged melt pool, the vapor pressure can splat in undesirable areas, thus resulting in a lack of material density and thickness in the region. A common factor found in the summation of the literature that was individually discussed and detailed (Section 1.3) is that energy densities within the 100–200 $J/m^3$ range tend to yield an optimal melt pool and relative material density, thus directly improving the mechanical and tribological properties as a byproduct of an enhanced surface finish. That being said, aside from these studies, there is a lack of literature that combines the relationship between as-built surface finish, tribology, and mechanical properties for AM Mg parts. Understanding this, this avenue of research may be useful to investigate in the near future

### 3.6. Nickel-Based Alloys

The last major metal-based material utilized in AM is nickel-based alloys. In general, nickel-based superalloys tend to garner the most interest in AM practices due to their unique properties in both elevated and ambient temperatures. In particular, the most impressive properties that this material exhibits would be their wear, oxidation, mechanical, and creep properties. Graybill et al. expand on this statement by listing the various applications that largely benefit from nickel-based superalloys in their comprehensive AM review of this metal [136]. Some examples listed are in the forms of turbochargers, nuclear reactors, and heat exchangers, which can be commonly found in environments such as steam power plants, aircraft, and rocket engines. Understanding that these applications generally operate in high-temperature conditions, it makes sense that nickel-based alloys are largely used. In the realm of AM, Lewandowski and Seifi summarized the common processes used with this material as the following: PBF, DED, EBM, and other specific powder/wire-based melting processes [31]. Lewandowski and Seifi support this through their vast literature review summarizing the various mechanical properties of AM'd metals. Additionally, the sources provided in Table 1 further support their claims.

Similar to aluminum alloys, there is a lack of research exploring the relationship between surface roughness and mechanical and tribological properties. In general, AM research for nickel-based alloys either focuses solely on mechanical properties, tribological properties, or surface roughness. Despite this gap in the research, this section will

attempt to form a relationship between these studies based on the evidence provided from earlier sections.

With regard to surface roughness, Hopskin and Mumtaz's experimentation on as-built Inconel 625 parts via SLM gives a clearer image of the relationship of laser-processing parameters and surface roughness [137]. Understanding the impacts of fatigue cracking due to excessive surface roughness (0.8 μm<), the authors sought to investigate the top and side roughness of the fabricated parts. From their results, it was shown that the top surface of the part tended to have a lower surface roughness compared to the side build. Interestingly enough, as the melt overlap was increased from the laser, the top section of the build decreased in surface roughness, whereas the side increased. The authors attribute this to surface tension forces that are exerted from the melt pool. If there is an increase in variation in thermal heat in the melt pool, then there is a greater likelihood of a balling effect occurring, similar to the results from the previous sections. Understanding that fatigue cracking is one property that affects both mechanical and tribological properties, it can be insinuated that the greater surface finish would promote a greater bonding of the metal particles, which, in turn, would directly improve the surface and bulk properties of the material.

For mechanical/tribological-based studies, Jia and Gu provide some great insight into the impacts of laser energy density to the hardness, and tribological properties of SLM Inconel 718 [85]. The variations in laser energy density are listed at 180, 275, 300 and 330 J/m through dividing the laser power (w) by the scan speed (mm/s). Interestingly, as laser energy density increased, there was an increase in densification of the Inconel 718 part as the dynamic viscosity of the melt pool was decreased, which, in turn, refined its grains. The authors also note that as the material became denser, the surface took a smoother finish. Relating to the performance of the smooth surfaces, there was an increase in consistent microhardness, which follows the trend of having a denser surface. In addition to this, the smoothest surface had the least amount of wear compared to all other laser densities. Although the surface roughness was not specifically measured, there is a definite connection between the surface finish and the mechanical/tribological properties of the material.

That being said, although there are no specific studies focusing on these three subjects, there is room for new research to be conducted on this topic as the optimization of processing parameters may greatly reduce the costs of post-processing techniques, that help to improve the quality of the part build.

**4. Optimization of Energy Conservation and Performance of Additively Manufactured Heat Exchangers from Refined As-Built Surface Quality**

In AM research, there are no existing publications which study the effects of as-built surface roughness on the tribological performance of heat exchangers and their implications for increased thermal efficiency and energy conservation. However, based on all of the information gathered on this point, it can be seen that, in general, there is a very close relationship between surface roughness and its impacts on the mechanical-related properties of AM components. In heat transfer research, there are some publications which do study the effects of surface roughness on the tribological and heat transfer performance of non-AM built heat-exchanging devices [22,138,139]. Geete and Pathak [138] found that surface roughness shared a relationship with the heat transferring rates of steel, aluminum, and copper pipes. In this work, computational observations were made with water and ammonia to a double-pipe counter flow heat exchanger with varying roughness. The exergy, entropy, and entransy were observed as the liquids passed through these pipes at a mass flow rate of 0.08 kg/s. As the relative surface roughness of the pipe was decreased, the heat transfer was increased as the smooth surface decreased the exergy, entropy generation, and entransy of the tested models. These decreasing components help to conserve the fluid's kinetic energy from the irreversibilities that typically occur in a system, thus increasing the heat-transferring performance and saving additional operating costs [140]. However, in Rayleigh–Bénard convection cells, the opposite case is true, where surface-textured surfaces have been shown to increase convective heat transferring. This was found in the work

of Tummers and Steunebrink [22], where arrayed copper cubes enhanced the turbulent kinetic energy of the ethanol- and water-based liquids increased due to a pluming effect from the surface.

From a tribological point of view, surface roughness also has an impact on the fretting fatigue and wear of tubing materials used in heat exchangers. Abbas et al. [139] studied this research using aluminum, copper, and stainless-steel tubes which were subjected to a constant vibrational force. It was observed that long periods of testing resulted in significant wear due to third-body abrasion from the initial wear loss of the tubes, suggesting that as fretting resistance improves, there is less of a chance that heat transferring tubes will fail. Over time, this may be very valuable to industrial power plants, as power and resources can be preserved. One way which this can be combatted is through the reduction in surface roughness, as shown in the work by Yue and Wahab [141]. As surface roughness is refined, the crack nucleation from the contacting asperities is decreased, preserving the material for long term-use. It should also be mentioned that in applications where tubes make fretting type of contacts, a decrease in surface roughness will also mitigate the heat loss of two contacting surfaces, as shown from Maisuria [142]. In an economical sense, the reduction in unnecessary heat losses will greatly help to improve the energy efficiency of heat-transferring devices.

This information suggests that surface roughness is a large contributor to the economic and heat-transferring performance of heat-exchanging devices. When applied to AM builds, control over the surface roughness allows for a new avenue of research which focuses on surface roughness in relation to heat transfer rates, and the wear performance of the built material over time, whether it is subjected to fluid flow or to the vibrational effects of surrounding tubes. AM also allows for the ability to control the texturing of the surface without the need for post-surface-refining techniques if optimized, which can be greatly beneficial to the complex geometric builds that AM offers compared to traditionally manufactured builds. With the current state of the literature on AM, control over surface roughness has been shown to increase the various mechanical properties of a build. Applying these principles to heat exchangers, it is to be expected that heat transfer rates can be improved due to a variety of phenomena, thus creating an incentive to explore this topic of research. If these processes are optimized, manufacturers will be able to conserve the large energy costs needed to run these processes, especially in large, industrial-scale power plants.

## 5. Conclusions and Future Directions

The popularization of AM technologies since their introduction has greatly impacted many industries due to the ability to create complex geometrical designs without the need for post-processing techniques. In heat-transfer research, AM is one technique which can be used to enhance the heat transfer rates of heat-exchanging devices and preserve the large sums of energy that are wasted from generated entropy and exergy. This review highlights the relationship of surface roughness to the mechanical, tribological, and heat transfer properties of metal-based AM heat exchangers. On the topic of optimizing as-built surface roughness, and as its effects on the energy conservation are novel, this review focused on highlighting the independent literature on surface quality optimization for AM builds, and their effects on the mechanical properties, surface roughness and performance of heat-exchanging devices. Given the absence of the literature covering these three topics, this review aimed to consolidate the existing literature on each individual topic and create a relationship between these processes.

First, the influence of surface roughness on the mechanical and tribological properties of AM metals was discussed. Some of the key findings in this analysis pertained to the influence of laser energy intensity on the final surface roughness of the substrate. Although AM processes may have a multitude of parameters that can impact the final build, the recent literature indicates that there is a range of laser intensities which optimize the surface roughness of the build, thus improving the bonding strength of the melted particles

and improving the densification of the build. These surface improvements have a direct correlation with mechanical and tribological performance. Viewing the impact of surface roughness on heat transfer rates, the amount of surface roughness in a build has been demonstrated to greatly impact the exergy, entransy, and entropy generation during a fluid flow process, indicating that finer surface finishes are desirable. Taking the fact that fretting wear and fatigue tends to occur in heat exchanging systems into consideration, the improvement in surface roughness from AM-processing parameters has tremendous value in the ever-growing market of AM-based heat exchangers. Creating a relationship between these processes, it is suggested that the control of surface roughness for heat exchangers will help to increase heat transfer rates and reduce the fretting wear presented from surrounding tubes.

In a general sense, it can be concluded that the current challenge in this field of research is the general lack of research. Although there is some research on the mechanical and tribological properties, the number of papers also focusing on improved surface qualities in conjunction with these properties for the energy-saving applications of heat transferring devices should be explored. One explanation of this could be the novelty of AM practices on a wide-spread industrial scale. Especially for heat-exchanging devices. This can be explained due to the material costs, the lack of production volumes, and the machining costs from AM. However, with AM becoming more widely utilized, these voids at both the micro- and macro-scale can be resolved in the near future. Based on this review, the authors suggest that additional research should be conducted in this field on a wide array of metals in order to fully link the relationship between as-built surface roughness and its impact on the energy conservation processes of heat-exchanging devices without the need for post-processing techniques.

**Author Contributions:** A.M.R.: conceptualization, methodology, formal analysis, and writing—original draft preparation; P.L.M.: conceptualization, methodology, writing—review and editing, and supervision; P.K.: conceptualization, methodology, and writing—review and editing. All authors have read and agreed to the published version of the manuscript.

**Funding:** This research was funded by NASA RRR CAN, award number: NV-80NSSC19M0172.

**Conflicts of Interest:** The authors declare no conflict of interest.

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
