# Peer review of "Tribological Properties of Additive Manufactured Materials for Energy Applications: A Review"

_processes, doi:10.3390/pr9010031_

Round 1
Reviewer 1 Report
It is a well written and comprehensive review related to tribological properties of additive manufactured materials for energy applications. Therefore, I recommend for publication in Processes. Prior to publication, I have a few minor comments as below, which the authors should address.
1. Lubricants can significantly influence the friction and wear characteristics as the authors mentioned. It is better the author can add a brief description about the lubrication mechanisms in this review. The lubrication mechanisms when lubrication film exists between two moving or sliding surfaces can be divided into hydrodynamic lubrication, solid lubrication, and boundary lubrication. Several recent related reviews [1-4] should be cited.
Hydrodynamic lubrication,
[1] Liu, H.; Liu, H.; Zhu, C.; Parker, R.G. Effects of lubrication on gear performance: A review. Mechanism and Machine Theory 2020, 145, 103701.
Solid lubrication
[2] Vazirisereshk, M.R.; Martini, A.; Strubbe, D.A.; Baykara, M.Z. Solid lubrication with MoS2: A review. Lubricants 2019, 7, 57.
Boundary lubrication
[3] Lin, W.; Klein, J. Control of surface forces through hydrated boundary layers. Current Opinion in Colloid & Interface Science 2019, 44, 94-106.
[4] Greenfield, M.L.; Ohtani, H. Friction and normal forces of model friction modifier additives in simulations of boundary lubrication. Molecular Physics 2019, 117, 3871-3883.
2. The authors should get permissions for Figure 1-4, and add the copyright permission information in the figure caption.
3. The Authors should pay attention to the format and typos of ms. I list just a few below as examples. Ex:It is better to use ‘Rsk’ instead of ‘Rsk’ (line 268 and 271), meanwhile please provide the full form before providing the abbreviated form in the main text. ‘Molybdenum’ should be ‘molybdenum’(line 269).
Reviewer 2 Report
This review paper provides a pleasant and informative reading on the surface integrity obtained by additive manufacturing, and how it influences the performance and service life of mechanical components. Very well written. The presentation and discussion are duly substantiated. However, figures and tables do not have an adequate formatting and must be corrected. The conclusions should also be improved.
Best Regards
